# ASVMR: Adaptive Support-Vector-Machine-Based Routing Protocol in the Underwater Acoustic Sensor Network for Smart Ocean

Shuyun Zhang [1], Huifang Chen [1,2,3,*] and Lei Xie [1,2]

1 College of Information Science and Electronic Engineering, Zhejiang University, Hangzhou 310027, China; 22131117@zju.edu.cn (S.Z.); xiel@zju.edu.cn (L.X.)
2 Zhejiang Provincial Key Laboratory of Information Processing, Communication and Networking, Hangzhou 310027, China
3 Zhoushan Ocean Research Center, Zhejiang University, Zhoushan 316021, China
* Correspondence: chenhf@zju.edu.cn

**Abstract:** The underwater acoustic sensor network (UASN) plays a crucial role in collecting real-time data from remote areas of the ocean. However, the deployment of UASN poses significant challenges due to the demanding environmental conditions and the considerable expenses associated with its implementation. Therefore, it is essential to design an appropriate routing protocol to effectively address the issues of packet delivery delay, routing void, and energy consumption. In this paper, an adaptive support vector machine (SVM)-based routing (ASVMR) protocol is proposed for the UASN to minimize end-to-end delay and prolong the network lifetime. The proposed protocol employs a distributed routing approach that dynamically optimizes the routing path in real time by considering four types of node state information. Moreover, the ASVMR protocol establishes a "routing vector" spanning from the current node to the sink node and selects a suitable pipe radius according to the packet delivery ratio (PDR). In addition, the ASVMR protocol incorporates future states of sensor nodes into the decision-making process, along with the adoption of a waiting time mechanism and routing void recovery mechanism. Extensive simulation results demonstrate that the proposed ASVMR protocol performs well in terms of the PDR, the hop count, the end-to-end delay, and the energy efficiency in dynamic underwater environments.

**Keywords:** underwater acoustic sensor network (UASN); intelligent routing protocol; support vector machine (SVM); packet delivery ratio (PDR)

## 1. Introduction

The ocean covers more than 70% of Earth's surface and provides valuable services to both humans and the environment, which makes the ocean monitoring crucial. Therefore, advanced technologies are required to monitor the assets effectively. In this respect, remote sensing offers an excellent opportunity to explore various oceanographic parameters through the utilization of archived, consistent, and multi-temporal datasets, all in a cost-effective manner [1,2]. However, traditional ocean remote sensing technologies are limited by several factors, such as weather, the limited coverage of sensing devices, and unreliable data transmission. In recent years, the underwater internet of things (UIoT), which can obtain real-time oceanic data and transmit it to the shore for further analysis and processing, is regarded as a new paradigm of ocean remote sensing.

Figure 1 illustrates the basic schematic of the UIoT, encompassing various modules for underwater sensing and transmission (underwater sensor nodes and surface nodes); underwater computing and transmission (autonomous underwater vehicles (AUVs)); and surface computing and transmission (surface base station (BS), surface ships, and surface nodes), as well as coastal control (seashore BS and seashore control center) [3–5].

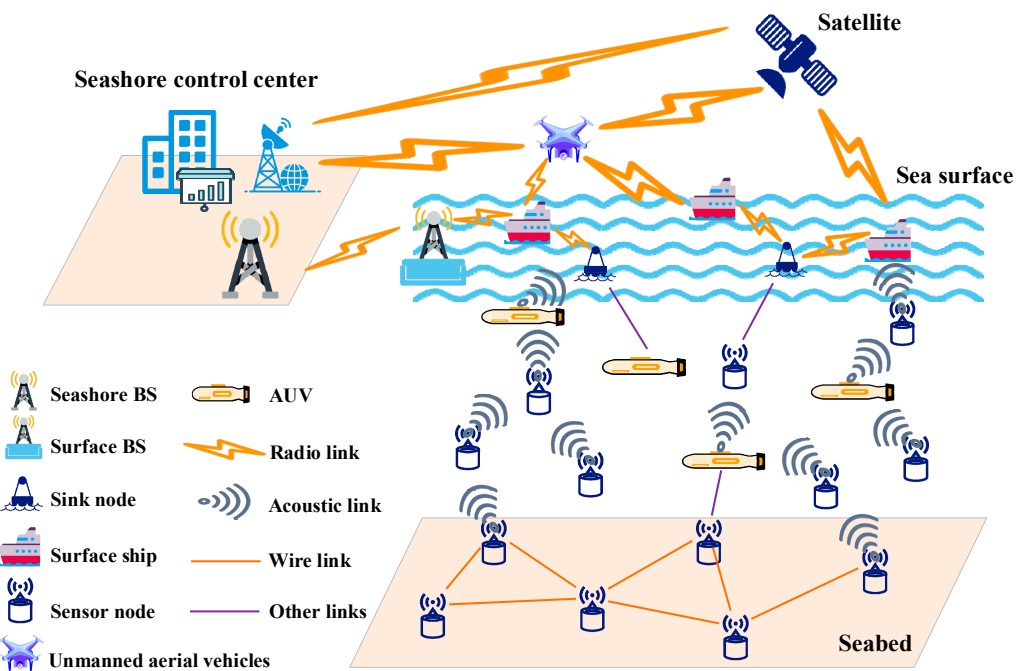

**Figure 1.** The schematic of the UIoT.

As the critical infrastructure in the UIoT, the underwater acoustic sensor network (UASN) collects data from remote areas of the ocean in real time, enabling the acquisition of rich and accurate ocean data [6]. By deploying numerous sensor nodes under water, the UASN plays a crucial role in improving remote sensing capability, understanding the complex dynamics of the ocean, and assessing the impact on the environment.

Deploying the UASN poses challenges due to the demanding environmental conditions and substantial deployment costs. First, the sound signal propagates much more slowly in the water, at approximately 1500 m/s, leading to noticeable latency in signal propagation. Second, some factors, such as water absorption, scattering, and underwater noise interference, limit the data transmission rate of the underwater acoustic signal. Third, the complexity and uncertainty of the underwater environment further increase the difficulty of underwater acoustic communication. Hence, underwater sensor nodes consume much more energy than their terrestrial counterparts when transmitting data of the same size [7]. In order to optimize energy utilization and enhance data transmission efficiency in the UASN, the design of an appropriate routing protocol is of paramount importance. This routing protocol should possess the flexibility to adapt to the dynamic underwater environment.

Over the past decade, many routing protocols specifically tailored for the UASN have been proposed [8–10]. Traditional routing protocols, which are typically based on a fixed routing table or static network topology, are not suitable for the UASN due to the dynamic nature of the underwater environment and the limitations of underwater acoustic signal transmission. The use of traditional routing protocols in the UASN can result in significant propagation latency, data transmission rate limitations, and increased energy consumption by underwater sensor nodes. These limitations bring about the routing void, long packet delay, and data packet loss, which can significantly degrade the network performance. Typically, two categories of conventional routing protocols exist in the UASN, namely the location-aware routing protocol and the depth-aware routing protocol.

For the location-aware routing protocol, it is assumed that the sensor nodes have the knowledge of the location information with the assistance of the node positioning technology. In the vector-based forwarding (VBF) protocol proposed in [11], data packets are forwarded in a virtual pipeline with a pre-defined radius. The virtual pipeline is specified by the routing vector from the location of the source node to the destination.

For the dense network, the VBF protocol effectively manages the network flooding area size to mitigate potential issues. However, in spare networks, the VBF protocol exhibits poor performance and fails to address routing voids. Hence, an enhanced VBF protocol, named hop-by-hop vector-based forwarding (HH-VBF), was proposed in [12]. The HH-VBF protocol employs the notion of a virtual routing pipe. This involves the utilization of an individual virtual pipeline for each forwarding node, and at each intermediate node, a directional choice is made based on its current location. Therefore, despite the limited number of neighboring nodes, the HH-VBF protocol can still find a data delivery path as long as a sensor node is available in the forwarding path within the communication range. However, the hop-by-hop nature introduces much more signaling overhead for the HH-VBF protocol. In [13], a geographic routing protocol and two topology control algorithms were introduced; they employed the greedy forwarding protocol as a foundation. When a data packet reaches a void node, the node has the capability to vertically adjust its position to establish connections with the non-void nodes and restore data forwarding. Nevertheless, the process of adjusting the sensor node's location consumes a substantial amount of energy. In [14], an adaptive location-based routing protocol (ALRP) was proposed. The ALRP introduces several key mechanisms to improve forwarding efficiency. First, it establishes a forwarding area to restrict the range of candidate forwarders. Second, it dynamically calculates the forwarding probability to minimize unnecessary redundancy in forwarding. Third, it adaptively determines the forwarding order based on the forwarding delay. To provide a decent performance, these algorithms need accurate 3 dimensional (3D) location information on the sensor nodes; this is difficult to obtain in the UASN [15].

For the depth-aware routing protocol, the routing decision is based solely on the depth information obtained from the sensor node's barometer. The depth-based routing (DBR) protocol, introduced in [16], represents the initial routing approach that utilizes the sensor node's depth for data forwarding in the UASN. Additionally, a holding time mechanism is implemented to facilitate the coordination of the forwarding candidates during transmission. However, in a sparse network, the greedy hop-by-hop forwarding may frequently encounter a communication void region, where the sensor node cannot find a next-hop node to deliver the data packet. The energy-efficient depth-based routing (EE-DBR) protocol, where both the residual energy and the depth of the sensor nodes are considered when selecting the next-hop node, was proposed in [17]. However, when the sensor nodes were deployed sparsely, the problem of the routing void was not resolved effectively. In [18], the distance vector-based opportunistic routing (DVOR) protocol was proposed. The DVOR protocol seeks the shortest routing path according to the hop count of the sensor nodes towards the destination. Additionally, a holding time mechanism was developed to regulate the scheduling of data packet forwarding. However, the DVOR protocol introduces an overhead in the UASN due to its reliance on periodic beacons for the dynamic establishment of routing paths. In [19], the adaptive power-controlled depth-based routing protocol (APCDBRP) was proposed to prolong the network lifetime. The protocol comprises two phases: route establishment and data transmission. Moreover, APCDBRP proposes a data protection and route reconstruction mechanism to address issues such as network topology changes. However, the power control and data protection mechanisms in APCDBRP introduce a certain level of end-to-end delay. In [20], a multi-layer cluster-based energy-efficient routing scheme was proposed. The proposed scheme encompasses three distinct stages. In the initial stage, the network is divided into multiple layers. Subsequently, in the second stage, the sensor nodes are organized into clusters within each layer. Finally, in the third phase, the data are efficiently forwarded towards the sink. To tackle the challenge of hotspots, a dynamic clustering approach is presented. This scheme can effectively balance network energy consumption and reduce end-to-end delay. A novel neighbor-based energy-efficient routing protocol was proposed in [21]. The protocol implementation comprises the deployment of random clusters and the relocation of nodes underwater. Notably, each node demonstrates the ability to detect, identify, and forward routing paths to the nearest neighboring node. These operations are facilitated

through the processes of route discovery and route maintenance. To enhance efficiency, the protocol employs a flooding mechanism, which effectively discovers the closest node by leveraging this approach. The depth-based routing methods use the greedy algorithm to forward data packets, where sensor nodes passively receive data packets. Although some methods have been used to limit the redundancy, the area around the forwarding nodes is still subject to flooding, resulting in energy waste. Additionally, due to the acoustic communication between the underwater sensor nodes, the transmission rate is limited, and excessive packet transmissions in the network easily lead to the failure of important packet forwarding.

With the development of artificial intelligence (AI), increasingly complex AI technologies are being used to design routing protocols [22]. Intelligent routing protocols have been proposed to address the challenges faced by traditional routing protocols in the UASN. These protocols can dynamically adapt to changes in the network environment, select the optimal path based on the real-time monitoring of network status and node changes, and optimize the energy consumption to reduce the routing void, minimize the packet delivery delay, and prolong the network life. In addition, intelligent routing protocols enhance the network security by selecting a more secure path.

In [23], a fusion algorithm of ant colony optimization algorithm (ACOA) and artificial fish swarm algorithm (AFSA) was proposed for the routing protocol in the UASN. An adaptive mechanism is used to combine the advantages of ACOA and AFSA. The method first uses the AFSA to calculate a set of globally optimal paths. To address the problem of insufficient precision in the optimal path calculation of the AFSA, a parallel ACOA is then employed to select the optimal path. However, the method is not suitable for resource-limited underwater nodes due to the high computational complexity. In [24], a Q-learning-based localization-free anypath routing (QLFR) protocol was proposed, where the calculation of the Q-value involves the simultaneous consideration of both the residual energy and the depth of the sensor nodes. Furthermore, a new holding time mechanism was developed for data packet forwarding, taking into consideration the priority of forwarding candidate nodes. Nevertheless, the intricate mechanism and substantial computational demand involved pose challenges for implementation in the underwater environment. In [25], a reinforcement learning-based opportunistic routing (RLOR) protocol was proposed by combining the advantages of opportunistic routing algorithms and reinforcement learning algorithms. In addition, the RLOR protocol incorporates a recovery mechanism that effectively enables data packets to bypass void areas and seamlessly continue their forwarding process, resulting in an improved packet delivery ratio (PDR), particularly in sparse networks. However, the RLOR protocol uses a specific value combination which cannot be dynamically adjusted to accommodate changes in the environment. In [26], the deep Q-network (DQN)-based energy and latency-aware routing (DQELR) protocol was proposed. The DQELR protocol uses DQN to train agents since the Q-learning-based methods are not suitable for environments with a large state space. Each data packet is defined as an agent, and the depth and residual energy are considered when designing the reward function. The DQELR protocol can extend the network lifetime, as well as satisfy the energy consumption and latency constraints. However, the additional cost resulting from the Q-learning-related information exchange is not addressed.

The routing methods mentioned above offer partial improvements in data transmission efficiency and energy consumption reduction. However, the challenges related to the high packet loss rate and prolonged end-to-end delay have not been addressed effectively. Furthermore, these methods lack the ability to adjust in real time and to recover the forwarding process when the data packets become trapped in routing voids. To address these issues, an adaptive support vector machine (SVM)-based routing (ASVMR) protocol is proposed for the UASN. In the proposed ASVMR protocol, the SVM is utilized to train the model for the selection of the relay node, where four factors are selected as features for the model. A reasonable routing pipe radius is chosen based on the PDR to minimize latency and extend the network lifetime. Moreover, a waiting time mechanism is designed

for the opportunity routing to improve the PDR. To deal with the transmission failure of the routing void, each sensor node can activate the recovery mechanism to bypass the void region and continue forwarding data packets. To the best of our knowledge, this study is the first attempt to employ the SVM model for the design of a routing protocol in the UASN. The main contributions of the paper can be summarized as follows.

1. Unlike the traditional routing protocols which select the relay node with a single parameter, the proposed ASVMR protocol employs a selection process to identify a group of forwarding candidates from neighboring nodes based on four factors, guaranteeing the optimal routing choice and enhancing the performance significantly.
2. The waiting time mechanism for opportunity routing is enhanced by incorporating the distance between sensor nodes, resulting in a reduction in both end-to-end delay and data packet loss.
3. A scheme for adaptive routing pipe radius is proposed to reduce unnecessary transmissions while also maintaining a high PDR.

The remainder of the paper is organized as follows. Section 2 contains the preliminaries, including the acoustic propagation model, the network model, and the SVM model. Section 3 presents the proposed ASVMR protocol in detail, and Section 4 elaborates on the design of corresponding routing protocol. In Section 5, the simulation results and discussions are given. Finally, Section 6 concludes the paper.

## 2. Preliminaries

In this section, the acoustic propagation model, the network model, and the SVM model are introduced.

### 2.1. Acoustic Propagation Model

Here, the Thorp model [27] is adopted for the underwater acoustic channel.

The attenuation of an underwater acoustic signal, characterized by frequency $f$ (in kHz) at transmission distance $l$ (in meter), can be described as

$$A(l,f) = l^k a(f)^l, \tag{1}$$

where $k$ represents the spreading coefficient, $a(f)$ is the absorption coefficient, and

$$10 \log a(f) = 0.11 \frac{f^2}{1+f^2} + 44 \frac{f^2}{4100+f^2} + 2.75 \times 10^{-4} f^2 + 0.003. \tag{2}$$

The energy expended by the sensor node for transmitting an $m$-bit data packet to another sensor node located at a distance $l$ ($l < l_{\max}$) can be expressed as

$$E_t(m,l) = mP_0 A(l,f), (E_t < E_{remain}), \tag{3}$$

where $P_0$ represents the minimum power necessary for the node to transmit data, $l_{\max}$ denotes the maximum communication distance of the sensor node, $E_{remain}$ is the remaining energy of the sensor node, and $E_{remain}$ represents the upper limit of $E_t$.

Similarly, the energy consumed by the sensor node to receive an $m$-bit data packet can be expressed as

$$E_r(m) = mP_r, \tag{4}$$

where $P_r$ is the reception coefficient.

Using the attenuation $A(l,f)$ can evaluate the signal-to-noise ratio (SNR) observed at a receiver over a distance $l$ when the transmitted signal is a tone of frequency $f$. Neglecting the directivity indices and losses other than the path loss, the narrow-band SNR ($SNR(l)$) is given by

$$SNR(l) = \frac{E_b / A(l,f)}{N_0}, \tag{5}$$

where $E_b$ is the average energy consumed to transmit one bit of data, and $N_0$ represents the noise power spectral density in an additive white Gaussian noise (AWGN) channel.

In this paper, binary phase shift keying (BPSK) modulation technology [28] is employed. When the propagation distance is $l$, the bit error rate can be computed as

$$p_e(l) = \frac{1}{2}\left(1 - \sqrt{\frac{SNR(l)}{1 + SNR(l)}}\right). \tag{6}$$

Consequently, the probability of successfully transmitting $m$-bit data is

$$p(m, l) = [1 - p_e(l)]^m. \tag{7}$$

### 2.2. Network Model

This paper considers a network architecture that includes multiple sink nodes [18], as depicted in Figure 2. The utilization of this architecture simplifies the practical deployment of the UASN. The network comprises a collection of sensor nodes, denoted as $\mathcal{SN}$, and sink nodes, denoted as $\mathcal{SK}$. The maximum transmission range of the sensor nodes is $R$. Thus, the network can be represented $\mathcal{N} = \mathcal{SN} \cup \mathcal{SK}$.

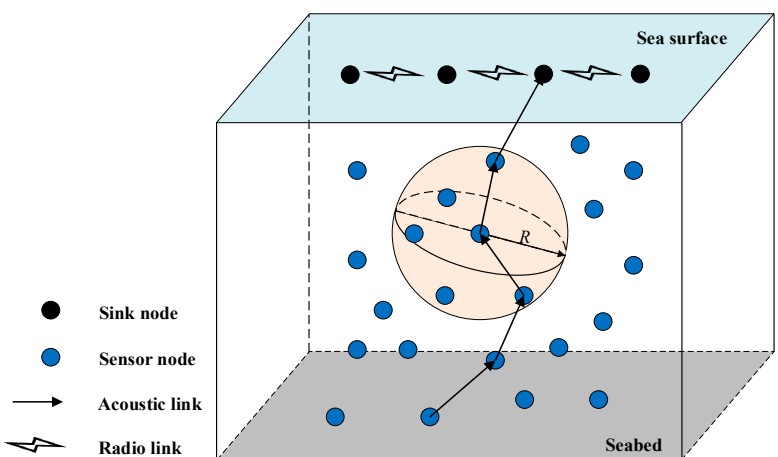

**Figure 2.** A network architecture of the UASN with multiple sink nodes.

In Figure 2, the 3D underwater area is randomly populated with sensor nodes that are equipped with sensing devices and acoustic modems. These nodes are responsible for performing observation and exploration tasks. Meanwhile, the sink nodes deployed on the water surface are equipped with both acoustic and radio frequency (RF) modems. The acoustic modem is used for the underwater communication, which includes the communication between the underwater sensor nodes and the communication between the underwater sensor nodes and the sink nodes. The RF modem is used for the surface communication, which includes the communication between the sink nodes and the communication between the sink nodes and the satellites. The underwater sensor nodes are responsible for gathering data from the monitoring areas and forwarding it to the sink nodes, which serve as the designated destinations for the underwater data packets. The sink nodes aggregate the sensory data and transmit them to the seashore control center for further processing and analysis by satellites.

In our assumption, we consider a successful delivery of a data packet to be when it reaches any sink node in the network.

### 2.3. SVM Model

The SVM model is a machine learning method based on statistical learning theory [29]; it can handle high-dimensional data, solve nonlinear problems, exhibit excellent generalization ability, and achieve high accuracy.

The core idea of the SVM model is to use a kernel function that satisfies the Mercer condition to replace nonlinear mapping. Hence, the sample points in the input space can be mapped to a high-dimensional feature space to facilitate linear separation, and an optimal hyperplane is constructed to approximate the ideal classification result. Hence, the SVM model is specifically used for small data samples and is a learning machine with optimal classification and generalization capability.

Given a set of training data samples, $\mathcal{F} = \{(x_1, y_1), (x_2, y_2), \cdots, (x_n, y_n)\}$, where $y_i \in \{-1, 1\}$, the hyperplane can be expressed as

$$\omega^T x + b = 0. \tag{8}$$

If the hyperplane $(\omega, b)$ can correctly classify the training data samples, we have

$$\begin{cases} \omega^T x_i + b \geq 1, & y_i = 1, \\ \omega^T x_i + b \leq -1, & y_i = -1. \end{cases} \tag{9}$$

An optimization problem supporting the vector machine model can be constructed as

$$\begin{aligned} &\min_{\omega, b} \tfrac{1}{2} \| \omega \|^2 + C \sum_{i=1}^{n} \xi_i, \\ &\text{s.t. } y_i (\omega^T x_i + b) \geq 1 - \xi_i, \ i = 1, 2, \cdots, n. \end{aligned} \tag{10}$$

where $\xi_i$ represents the degree of misclassification for sample $i$. In order to control the degree of misclassification, an error penalty factor $C$ is introduced.

As a convex quadratic programming problem, the problem formulated in (10) can be solved by transforming to the dual problem using the Lagrange multiplier method. That is,

$$L(\omega, b, \xi, \alpha, \beta) = \frac{1}{2} \| \omega \|^2 + C \sum_{i=1}^{n} \xi_i - \sum_{i=1}^{n} \alpha_i \times [y_i(\omega x_i + b) - 1 + \xi_i] - \sum_{i=1}^{n} \beta_i \xi_i, \tag{11}$$

where $\alpha_i$ and $\beta_i$ are Lagrange multipliers, $\alpha_i \geq 0$ and $\beta_i \geq 0$.

The dual problem of (11) is obtained as

$$\begin{aligned} &\max_{\alpha} \sum_{i=1}^{n} \alpha_i - \tfrac{1}{2} \sum_{i=1}^{n} \sum_{j=1}^{n} \alpha_i \alpha_j y_i y_j k(x_i, x_j), \\ &\text{s.t. } \sum_{i=1}^{n} \alpha_i y_i = 0, \ 0 \leq \alpha_i \leq C, \ i = 1, 2, \cdots, n. \end{aligned} \tag{12}$$

The decision function, denoted as $f(x)$, can be expressed as

$$f(x) = \omega^T x + b = \sum_{i=1}^{n} \alpha_i y_i k(x_i, x) + b, \tag{13}$$

where $k(\cdot, \cdot)$, the kernel function, can be a linear function, a radial basis function (RBF), or a polynomial function.

## 3. Proposed ASVMR Protocol

In this section, the proposed ASVMR protocol is presented in detail. The framework of the SVM adapted for routing is introduced first. Then, a detailed description of the proposed ASVMR protocol, including the determination of the next hop, the dynamic timer, the adaptive pipe radius scheme, and the recovery mechanism, is presented.

### 3.1. The Framework of SVM

To minimize latency and extend the overall lifespan of the network, four factors, namely the ratio of the depth difference and the maximum communication range, the ratio of the depth difference and the distance of the sensor nodes, the residual energy function, and the neighboring node function, are selected as features for training the SVM model for routing. These factors collectively form a four-dimensional sample, represented by $x_k$.

Suppose that node $n_i$ sends a data packet to node $n_j$, $(x_i, y_i, z_i)$ and $(x_j, y_j, z_j)$ represent the positions of nodes $n_i$ and $n_j$, respectively. The ratio of the depth difference and the maximum communication range $R$ can be defined as

$$drRatio(n_i, n_j) = \frac{z_j - z_i}{R}. \tag{14}$$

The ratio of the depth difference and the distance between nodes $n_i$ and $n_j$ can be defined as

$$ddRatio(n_i, n_j) = \frac{z_j - z_i}{\sqrt{(x_j - x_i)^2 + (y_j - y_i)^2 + (z_j - z_i)^2}}. \tag{15}$$

The residual energy function at node $n_j$ can be defined as

$$f_e(n_j) = \frac{e_{res}(n_j)}{e_{ini}(n_j)}, \tag{16}$$

where $e_{ini}(n_j)$ and $e_{res}(n_j)$ denote the initial and residual energy of node $n_j$, respectively. The neighboring node function at node $n_j$ can be defined as

$$f_n(n_j) = \frac{nei(n_j)}{nei_{max}}, \tag{17}$$

where $nei(n_j)$ denotes the number of neighboring nodes connected to node $n_j$, and $nei_{max}$ is the maximum number of neighboring nodes among all the sensor nodes in the network.

Therefore, the sample $k$ can be expressed as $x_k = \{drRatio, ddRatio, f_e, f_n\}$.

Here, we employed a portion of the publicly available ASUNA dataset [30], which consists of 11,000 sample groups, including 6561 positive and 4439 negative samples. The partial training samples are listed in Table 1.

**Table 1.** Partial training samples.

| drRatio $x_1 \in (0, 1]$ | ddRatio $x_2 \in (0, 1]$ | $f_e$ $x_3 \in (0, 1]$ | $f_n$ $x_4 \in [0, 1]$ | Label $y \in \{-1, 1\}$ |
|---|---|---|---|---|
| 0.6 | 0.6 | 0.9 | 0 | $-1$ |
| 0.2 | 0.1 | 0.7 | 0.6 | $-1$ |
| 0.7 | 0.4 | 0.5 | 0.2 | 1 |
| 0.4 | 0.8 | 0.4 | 0.4 | 1 |

Here, the value of $drRatio$, $ddRatio$, and $f_e$ is set within $(0, 1]$, and the value of $f_n$ is set within $[0, 1]$. The value of $y$ is determined according to the result provided by ASUNA and manual judgment. For example, in the first sample of Table 1 the node has no neighboring nodes with $x_4 = 0$, and thus, the label $y$ is set to $-1$. We randomly selected 2500 samples for the training of the model, normalized them, and employed 5-fold cross-validation with an RBF kernel function. The remaining 8500 test samples were utilized to evaluate the trained SVM model. The test results showed that out of 8499 samples, the predicted values of the labels $y$, calculated by the SVM model, matched the true values. Therefore, the test accuracy achieved 99.988%.

### 3.2. The Determination of Next Hop

In our selection process for the candidate forwarding set from the neighboring nodes, we take into account both the node depth and the pipe radius of the routing vector, ensuring a comprehensive consideration of these factors. The set of sensor nodes in the UASN is defined as $S\mathcal{N} = \{n_1, n_2, \cdots, n_m\}$, where $m$ is the number of sensor nodes.

First, the sensor nodes in the routing vector and the above node $n_i$ are grouped into the candidate forwarding set of node $n_i$. The illustration of the candidate forwarding set

selection is depicted in Figure 3, where $\mathcal{N}_{\text{nei},i}(t)$ represents the set of neighboring nodes of node $n_i$ at time $t$ and $\mathcal{S}_{\text{above},i}(t)$ represents the candidate forwarding set selected for node $n_i$ at time $t$. The shaded rectangle is the routing pipe.

Second, node $n_i$ acquires the state values of the sensor nodes in the candidate forwarding set and incorporates these values into the SVM model to obtain the one-hop decision value $v_t$ at time $t$ in (13). In order to reflect the influence of future states on the current state, the decision value at time $t$ is defined as

$$V_t = v_t + \gamma v_{t+1} + \gamma^2 v_{t+2} + \cdots = \sum_{i=0}^{\infty} \gamma^i v_{t+i}. \tag{18}$$

where $\gamma$ denotes the discount factor.

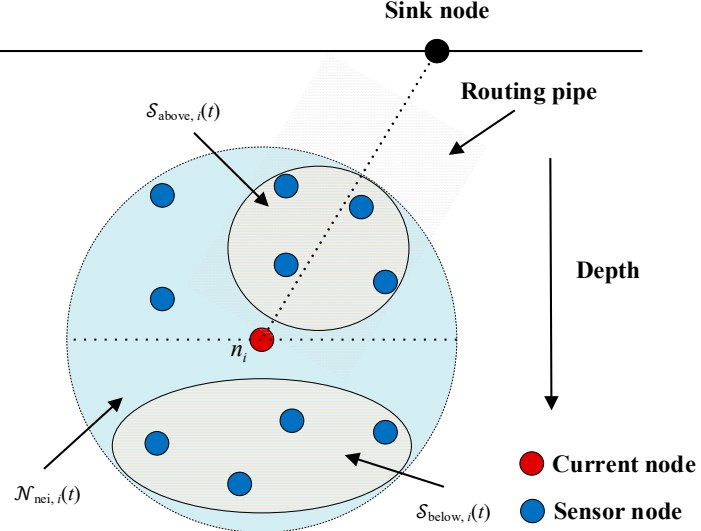

**Figure 3.** Illustration of candidate forwarding set selection.

To reduce the computational complexity, we only compute the value $V_t$ for $\gamma$ raised to the first power, and the node with the maximum value of $V_t$ is chosen as the next hop. It is worth noticing that the four factors of a node are constantly changing due to the continuous movement underwater and the energy depletion. Therefore, $V_t$ varies with time $t$, and the latest feature space information is utilized for each calculation of $V_t$.

### 3.3. A Dynamic Timer

The preceding section primarily focuses on selecting the most suitable next-hop node. However, in underwater environments, communication via single-path transmission is unreliable. To enhance the PDR, this model adopts an opportunistic routing approach. Furthermore, a dynamic timer is employed to correlate the waiting time with the distance between the current node and the optimal next-hop node, ensuring that the data packets can be transmitted to the previously selected optimal next-hop node.

Opportunistic routing involves a node initially forwarding data packets to a group of potential nodes, each of which retains a copy of the data packets [31]. Subsequently, each potential node can set its own timer to determine how long it will keep the copy. When the timer expires, the respective node becomes the designated relay node, and the other potential nodes can observe this behavior and discard their copies. This mechanism improves the reliability of data transmission, reduces unnecessary redundancy, and helps conserve energy. Nevertheless, the inclusion of timers introduces additional latency into the end-to-end communication.

To further decrease the end-to-end delay, an adaptive timer setting based on node distance is addressed in this paper. Specifically, closer nodes have shorter waiting times, allowing them to forward data packets more swiftly and reducing the end-to-end delay.

Figure 4 depicts a scenario where node $n_i$ has a set of neighboring nodes and where node $n_j$ has the maximum decision value. In this case, node $n_i$ selects node $n_j$ as the next-hop node and transmits the data packet to it, along with the ID and position of node $n_j$. Upon receiving the packet, node $n_j$ immediately forwards it, while the other nodes store a copy and calculate their distance from node $n_j$. If this distance is less than the maximum communication range $R$, the node sets a timer. If a forwarding packet from node $n_j$ is not received before the timer expires, it forwards the stored replica in the hope of reaching node $n_j$. The variable $W$ represents the routing pipe radius, and the size of the candidate forwarding set can be dynamically adjusted based on the routing pipe radius. The maximum value of $W$ is $R$.

Assuming that node $n_k$ is within the maximum communication range of both node $n_i$ and node $n_j$, the waiting time of node $n_k$ can be constructed as

$$T_{\text{wait}}(k) = \frac{d_{ij} + d_{jk} - d_{ik}}{v} + rand\left(0, \frac{R}{v}\right). \tag{19}$$

where $d_{ij}$, $d_{jk}$, and $d_{ik}$ represent the distance between node $n_i$ and node $n_j$; node $n_j$ and node $n_k$; and node $n_i$ and node $n_k$, respectively; $v$ is the sound speed in water; function $rand(a, b)$ generates a random real number within range $(a, b)$. In (19), the first term reflects the waiting time required for node $n_k$ to receive the data packet from node $n_i$ and then from node $n_j$. The second term represents a random waiting time set to avoid the occurrence of collisions between nodes with the same waiting time.

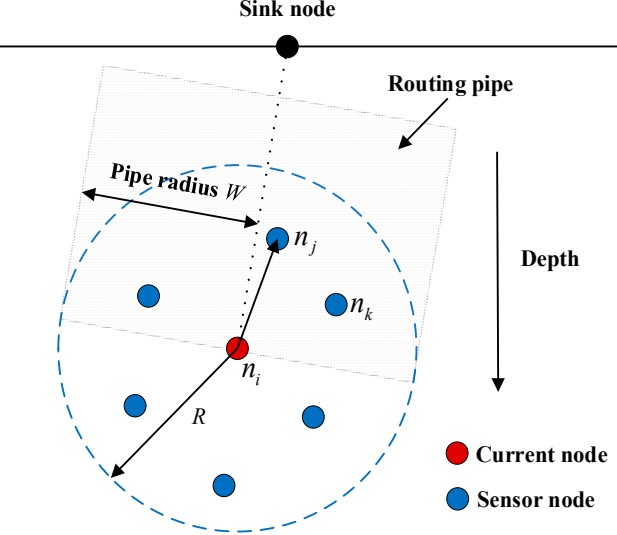

**Figure 4.** Neighbor selection, forwarding, and waiting time calculation.

### 3.4. Adaptive Pipe Radius Scheme

In practice, it is typical to select a sufficiently large routing pipe radius to increase the number of candidates forwarding nodes and improve the PDR. However, a larger radius also enables more sensor nodes with comparable waiting times to forward the same data packet, leading to redundant transmissions and energy waste. To enhance energy efficiency, it is necessary to impose additional restrictions on the data packet transmissions during the routing process.

However, if the data packet transmissions are suppressed excessively, it will result in a reduction in the PDR. The PDR is a measure of transmission reliability. In a sparse network, where it is essential to increase the PDR, the pipeline imposes fewer restrictions on the participating sensor nodes. Conversely, in a dense network, to minimize unnecessary energy consumption, the pipeline imposes limitations on the sensor nodes involved in routing. Therefore, to enhance the energy efficiency and maintain high transmission reliability, we propose an adaptive pipe radius scheme.

First, the routing pipe radius is initialized as the maximum value $R$. To balance transmission reliability and energy consumption, a PDR threshold is utilized, which can be tailored to suit the specific requirements of the practical application scenario in the UASN.

Second, the source attaches the number of generated data packets to the transmitted data packet during the data packet transmission phase. After the data packet is received, the sink node determines the PDR by dividing the quantity of data packets successfully delivered by the overall count of generated data packets.

If the PDR exceeds the threshold, the routing pipe radius will be reduced during the subsequent transmission to improve energy efficiency. If the PDR falls below the threshold, the sink node initiates a broadcast message to expand the routing pipe radius, and the source will attach the new routing pipe radius to the transmitted data packet. This will result in an increase in the pipe radius of eligible forwarders during the next transmission round, which will improve the delivery ratio.

The proposed adaptive pipe radius scheme is illustrated in Algorithm 1.

---

**Algorithm 1:** Adaptive Pipe Radius Scheme

---

$R$ is the communication range of nodes. $P_{\text{gen}}$ is the cumulative count of data packets generated. $P_{\text{rec}}$ denotes the number of data packets that have been successfully received. $PDR$ is the current PDR. $PDR_{\text{th}}$ is the predefined threshold of PDR.

  1:    Initialize the routing pipe radius to $R$
  2:   **while** the packet transmission phase is ongoing **do**
  3:      Commence a fresh iteration of data packet transmission
  4:      Attach $P_{\text{gen}}$ to the transmitted data packet at the source
  5:      Calculate the PDR at the sink node using $PDR = \frac{P_{\text{rec}}}{P_{\text{gen}}}$
  6:      **if** $PDR > PDR_{\text{th}}$ **then**
  7:        Decrease the routing pipe radius during the next transmission round
  8:      **else**
  9:        Increase the routing pipe radius during the next transmission round
10:      **end if**
11:  **end while**

---

### 3.5. Recovery Mechanism

In situations where a node is unable to find neighboring nodes located closer to the sink node, a void node emerges [32]. During data transmission, selecting a void node as the next hop will result in the data packet loss, which depletes energy and diminishes the data transmission efficiency. To address this issue, the proposed routing method avoids selecting void nodes to proactively trigger the recovery mode when encountering a routing void.

First, the number of neighboring nodes is taken as a dimension in the feature space for training. The likelihood of selecting a node as the next hop increases if it has a higher number of neighboring nodes. However, the approach does not completely eliminate the occurrence of routing voids, and there is still a possibility that certain void nodes may be selected. Thus, a recovery mode is incorporated that enables void nodes to locate a suitable next hop for forwarding the data downward, bypassing the void area effectively.

In the recovery mode, the candidate forwarding set for node $n_i$ is composed of the nodes below and inside its pipeline, denoted as $\mathcal{S}_{\text{below},i}(t)$, as illustrated in Figure 5. Then, the next hop is determined by the value of $V_t$. Upon successfully transmitting the data packet to a non-void node, the void node concludes the recovery mode and resumes its role in routing data packets towards the water surface. During this operational state, the node maintains a record of the previous hop node's ID to avoid any occurrence of routing loops.

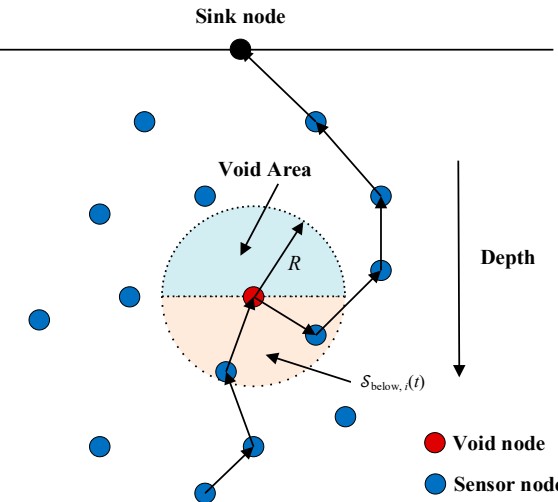

**Figure 5.** Illustration of recovery mode.

## 4. The Design of Routing Protocol

In this section, an elaborate outline of the devised routing protocol design is presented, including the packet structure, the exchange of node status knowledge, and the forwarding of data packets.

### 4.1. The Packet Structure

Figure 6 illustrates the packet structure used in the network; it comprises a header composed of the packet identification, the routing information, and the node status information.

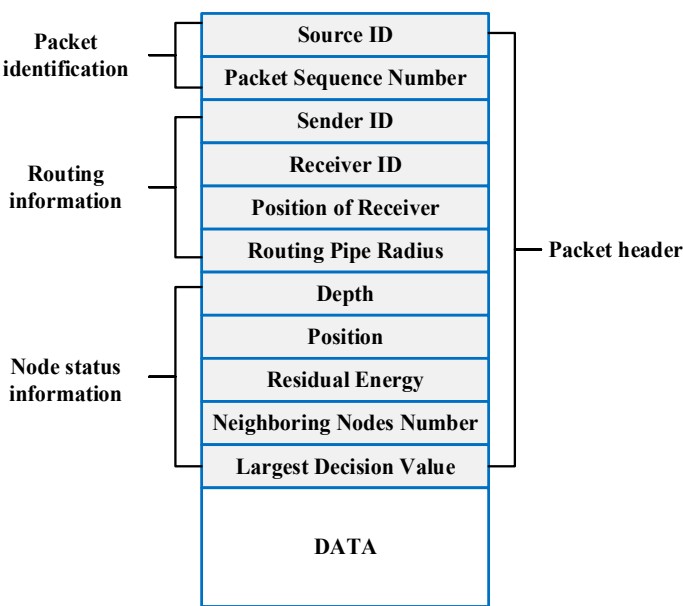

**Figure 6.** The structure of packet delivery.

The packet identification fields include:

(1)   Source ID, identifying the source node.

(2)   Packet sequence number, providing a unique identifier for the packet.

These fields are node-specific and utilized to differentiate data packets during data forwarding, remaining constant throughout the packet's lifetime.

Routing information is used to determine the routing pipe radius, select the next hop, and assist the forwarding candidates in the transmitted data packets. The routing information comprises the following fields:

(1)    Sender ID, identifying the current node.
(2)    Receiver ID, identifying the optimal next hop.
(3)    Position of receiver, providing the 3D coordinates of the optimal next hop.
(4)    Routing pipe radius, specifying the routing pipe radius.

The routing pipe radius size controls the number of forwarding candidates, as discussed previously. The sink node determines the size through the calculation of the PDR.

Each node must embed its status information into the following fields before sending a data packet.

(1)    Depth, providing the depth information of the current node.
(2)    Position, providing the 3D coordinates of the current node.
(3)    Residual energy, providing the remaining energy of the current node.
(4)    Neighboring nodes number, indicating the number of neighboring nodes of the current node.
(5)    Largest decision value, providing the largest decision value among the neighboring nodes of the current node.

Upon the reception of a data packet, every node extracts the relevant fields from the packet header and refreshes its neighboring information with the most up-to-date routing details. This process aids the nodes in making informed routing decisions that optimize their routing paths.

Apart from the packet header, there is an optional data field that can be included. This field carries the message intended for transmission to the destination. If the data field is not present, the packet serves the sole purpose of exchanging routing information, as further detailed in the subsequent subsection.

### 4.2. Node Status Knowledge Exchange

To optimize the routing decision-making process, it is necessary for all sensor nodes to possess their neighboring nodes' status information to calculate the decision values using an SVM model. The proposed routing protocol utilizes two approaches for exchanging node status information.

- Simultaneous Exchange with Data Packet Transmission: In this paper, the sender's status information is appended to the data packet header prior to its transmission. Consequently, a node can obtain its neighboring nodes' status information from the incoming data packets.
- Use of Hello Packets Containing Node Status Knowledge: Each node in the UASN periodically broadcasts a Hello packet, used solely for exchanging status knowledge. These broadcasts complement the approach whereby node status knowledge is exchanged. As each node can obtain the status knowledge of the neighboring node(s) from data packet transmissions, special control packets do not need to be used. Therefore, the broadcast period of the Hello packet can be configured with an adequate duration to eliminate the overhead.

### 4.3. Data Packet Forwarding

This part discusses the procedure of data packet forwarding in the proposed ASVMR protocol, as summarized in Algorithm 2.

Before initiating the transmission of a data packet, the sender conducts a preliminary examination for any routing voids. If present, it enters a recovery mode and forms a candidate forwarding set by selecting neighboring nodes that simultaneously possess two characteristics, namely nodes below the sender and nodes in the routing pipe. Alternatively, if routing holes are absent, a candidate forwarding set is formed by the neighboring

nodes that simultaneously possess two characteristics, namely nodes above the sender and nodes in the routing pipe.

Next, the sender calculates decision values for each potential forwarding set based on the obtained status information. The node with the maximum decision value is chosen as the next hop, and the remaining nodes in the candidate forwarding set assist in routing the data packet towards the most favorable next hop. The routing pipe radius controls the number of nodes in the candidate forwarding set. Before transmitting the data packet, the node modifies the packet header by incorporating its own status information and the details of the next hop.

Upon reception of a data packet, a node retrieves the sender's status information from the packet header and updates the relevant neighboring information, regardless of its role as a qualified forwarder.

Then, the node verifies whether it has previously forwarded the same data packet. If it has, the node directly discards the data packet; otherwise, it checks whether it is the receiver. If the node is the receiver, it forwards the packet following the above procedure. If the node is not the receiver, it calculates its distance from the receiver. If the distance exceeds the communication range, the node discards the data packet. Otherwise, the node initiates a waiting time. While waiting, if a node intercepts the same data packet, it refrains from forwarding it since another node has already taken the responsibility of transmission. If not, the node proceeds with the transmission of the data packet once the waiting time period has expired.

Furthermore, the proposed method employs an online and interactive training process. As previously stated, in every round of packet transmission, the sender calculates decision values for each candidate forwarding set using the acquired status knowledge before sending a data packet.

---

**Algorithm 2:** Data Packet Forwarding

---

*Packet* represents the data packet. $n_i$ represents the node that is presently receiving the data packet. $n_j$ is the receiver in the header of *Packet*. $V_{n_x}$ is the decision value of node $n_x$. $\mathcal{C}_i \in \{\mathcal{S}_{\text{below},i}, \mathcal{S}_{\text{above},i}\}$ is the candidate forwarding set of $n_i$. $d_{ij}$ is the distance between node $n_i$ and node $n_j$. $R$ is the communication range of $n_i$. $T_{\text{wait}}(i)$ is the waiting time to hold the data packet at node $n_i$.

 1:   On hearing *Packet*
 2:   Get the information from the header of *Packet*
 3:   **if** $n_i$ has forwarded *Packet* **then**
 4:       Drop *Packet*
 5:   **else if** $n_i == n_j$ **then**
 6:       Calculate $V_{n_x}$ for $n_x \in \mathcal{C}_i$
 7:       Choose the maximum $V_{n_x}$
 8:       Update the header of *Packet*
 9:       Send *Packet* immediately
10:   **else**
11:       Calculate $d_{ij}$
12:       **if** $d_{ij} > R$ **then**
13:           Drop *Packet*
14:       **else**
15:           Calculate $T_{\text{wait}}(i)$
16:           **if** $n_i$ overhears *Packet* during $T_{\text{wait}}(i)$ **then**
17:               Drop *Packet*
18:           **else**
19:               Revise the *Packet* header with updated information
20:               Send *Packet* when $T_{\text{wait}}(i)$ epires
21:           **end if**
22:       **end if**
23:   **end if**

---

## 5. Simulation Results and Discussion

The performance of the proposed ASVMR protocol is evaluated by computer simulations.

### 5.1. Simulation Setup

In the UASN, each sensor node in the network has a unique ID and limited energy. Moreover, it has the knowledge about the locations of sink nodes, the sender (via the packet header), one-hop neighboring nodes, and its own location through underwater localization algorithms [33]. The sensor nodes are uniformly distributed within a 3D area measuring 500 m × 500 m × 500 m. The sensor nodes only receive information from one-hop neighboring nodes. The sink nodes, which have unlimited energy and the ability to communicate using RF and acoustical modes, are fixed on the water surface. The source nodes, which have the ability to move horizontally, are deployed at the bottom. The locations of sensor nodes are illustrated in Figure 7. The parameters in the simulations are listed in Table 2.

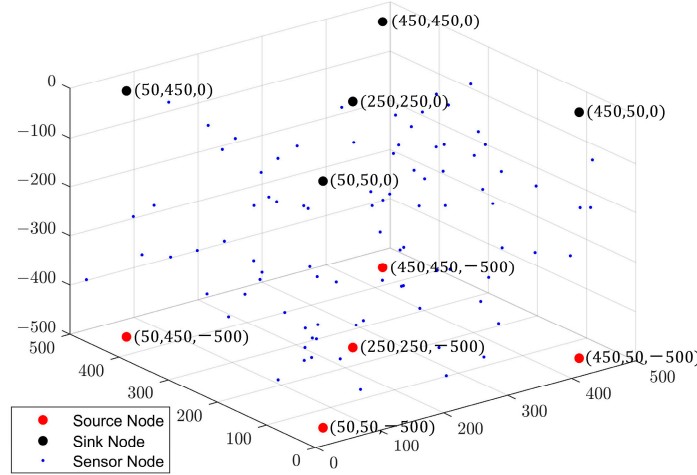

**Figure 7.** Placement of sensor nodes in the UASN.

**Table 2.** Parameter settings in simulations.

| Parameter | Value |
| --- | --- |
| Simulation time | 5000 s |
| The number of sink nodes | 5 |
| The number of source nodes | 5 |
| Sound speed | 1500 m/s |
| Communication radius | 150 m |
| Carrier frequency | 25 kHz |
| Data generation rate | 1 packet/s |
| Transmission rate | 10 kbps |
| Power of transmission | 2 W |
| Power of reception | 0.1 W |
| Power of idle | 10 mW |
| Number of sensor nodes | 100~500 |
| Node moving velocity | 0~2 m/s |
| Discount factor $\gamma$ | 0.1~1 |
| Routing pipe radius $W$ | $R/8$~$R$ |

In addition, four quantitative metrics are adopted in the simulations to evaluate the performance of the proposed ASVMR protocol.

(1) The PDR: The proportion of data packets successfully received by the sink node compared to the total number of data packets transmitted by the source node.

(2) The hop count: The average number of intermediate nodes traversed by a data packet from the source node to the sink node.

(3)   The end-to-end delay: The average latency of a data packet from the moment it is transmitted by the source node to its reception at the sink node.

(4)   The energy tax: The average amount of energy expended by each node in forwarding a data packet towards the sink node.

*5.2. Performance Comparison*

To comprehensively evaluate the performance of the proposed ASVMR protocol, four routing protocols are selected for comparison, namely (1) the DBR protocol [16], (2) the HH-VBF routing protocol [12], (3) the flooding routing protocol [34], and (4) the RLOR protocol [25]. Here, the DBR and HH-VBF protocols are the representative depth-aware and location-aware routing protocols, respectively. Additionally, to better demonstrate the performance of the proposed ASVMR protocol, the flooding routing protocol and the RLOR protocol in underwater networks are also included in the comparison. Table 3 provides a comparison of these five routing protocols.

**Table 3.** Comparison of routing protocols.

| Protocol | Year | Advantage | Disadvantage |
|---|---|---|---|
| DBR [16] | 2008 | Only deep information is required, with high PDR | Easy to cause energy hole problem and consume a lot of energy |
| HH-VBF [12] | 2007 | Robustness improved | Significant signaling overhead |
| Flooding [34] | 2017 | Simple, link reliability | Poor performance, high overhead |
| RLOR [25] | 2021 | High energy efficiency | Computational complexity |
| ASVMR | 2023 | Minimizes end-to-end delay and prolongs the network lifetime | Node location information needed |

First, the impact of node density on the performance of the five routing protocols is simulated and compared.

Figure 8 shows the impact of node density on the performance of the five routing protocols, where the node density changes as the number of sensor nodes varies from 100 to 500; the moving velocity of sensor nodes is set as 0; the discount factor is set as 0.8; and the routing pipe radius is set as *R*. The PDR, the hop count, the end-to-end delay, and the energy tax are shown in Figure 8a–d, respectively.

From Figure 8a, we observe that as the node density increases, the PDR of the five routing protocols increases gradually. This is due to the reduction in the number of void regions resulting from the deployment of nodes from sparse to dense, which allows more nodes to participate in packet forwarding and mitigates packet loss. Moreover, the proposed ASVMR protocol outperforms the others since it utilizes opportunistic routing and a recovery mechanism to enhance the success probability at each hop. Additionally, in a sparse network, the ASVMR protocol expands its pipeline radius to its maximum value (i.e., as the communication range) to eliminate constraints on the pipeline.

From Figure 8b, one finds that as the node density increases, the hop count decreases progressively across the five routing protocols. In a sparse network, sensor nodes may not cover the shortest path between the source and the sink node, resulting in multiple triggering of the recovery mode and a high average hop count. Nonetheless, in a densely populated network, the presence of additional nodes leads to a reduced number of void nodes and an increased likelihood of node coverage along the shortest path, resulting in a decreased hop count. In a sparse network, the RLOR protocol exhibits a higher hop count compared to the DBR protocol due to a smaller number of nodes. This limitation hinders reinforcement learning from acquiring additional valuable information, thereby leading to a degradation in performance. Additionally, the hop count of the proposed ASVMR protocol outperforms that of the others since it integrates an adaptive pipe radius and the SVM mechanism to identify the globally optimal next hop.

The proposed ASVMR protocol demonstrates effective performance in terms of hop count, which results in low end-to-end delay and energy consumption, as illustrated in Figure 8c,d, respectively. By defining a pipeline, the ASVMR protocol ensures that data are exclusively transmitted along the shortest path from the current sensor node to the sink node. Hence, the unnecessary energy consumption caused by data diffusion or dispersion in multiple directions is mitigated. In particular, in a dense network scenario, the pipeline radius is reduced to decrease the energy consumption while achieving the desired PDR. The incorporation of the SVM model in the ASVMR protocol facilitates the selection of sensor nodes with higher remaining energy. This contributes to a more balanced distribution of data traffic and prevents premature node failures caused by excessive energy consumption. Moreover, the ASVMR protocol employs the SVM model to determine the minimum hop count, which reduces the end-to-end delay and energy consumption.

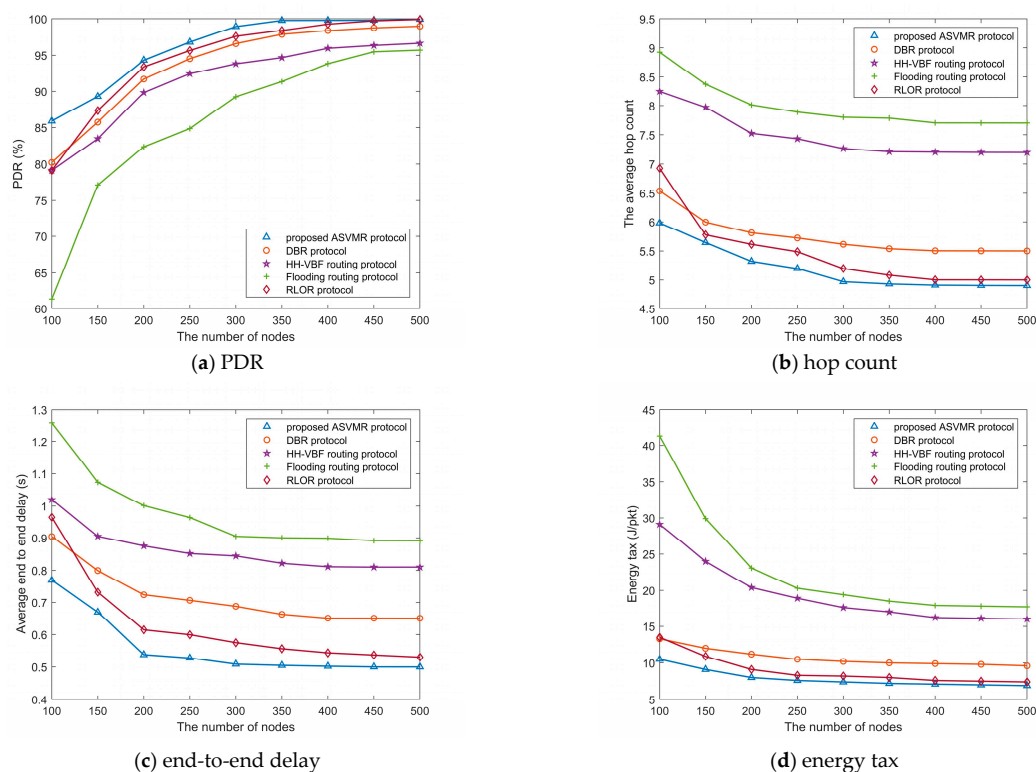

**Figure 8.** Impact of the node density on the performance.

Second, the impact of node mobility on the performance of five routing protocols is simulated and compared.

Figure 9 shows the impact of node mobility on the performance of five routing protocols, where the number of sensor nodes is set as 150; the node moving velocity changes from 0 m/s to 2 m/s; the discount factor is set as 0.8; and the routing pipe radius is set as *R*. The PDR, the hop count, the end-to-end delay, and the energy tax are shown in Figure 9a–d, respectively.

From Figure 9a, we observe that as the node moving velocity increases, the PDR of the five routing protocols increases slightly. Therefore, in a sparse network, the mobility of the sensor nodes has only a slight influence on the PDR. This phenomenon occurs because the network topology changes rapidly when the node velocity increases, resulting in a rapid coverage of the void region. However, as the network becomes denser, the emergence of void regions decreases.

From Figure 9b, one finds that as the node moving velocity increases, the hop count of the five routing protocols changes slightly. Therefore, the moving velocity of the sensor nodes has a negligible influence on the hop count.

From Figure 9c,d, we observe a slight decrease in the end-to-end delay and the energy consumption of the five routing protocols as the node moving velocity increases. This is because the node mobility can improve network coverage, which slightly enhances the data packet transmission efficiency in the network.

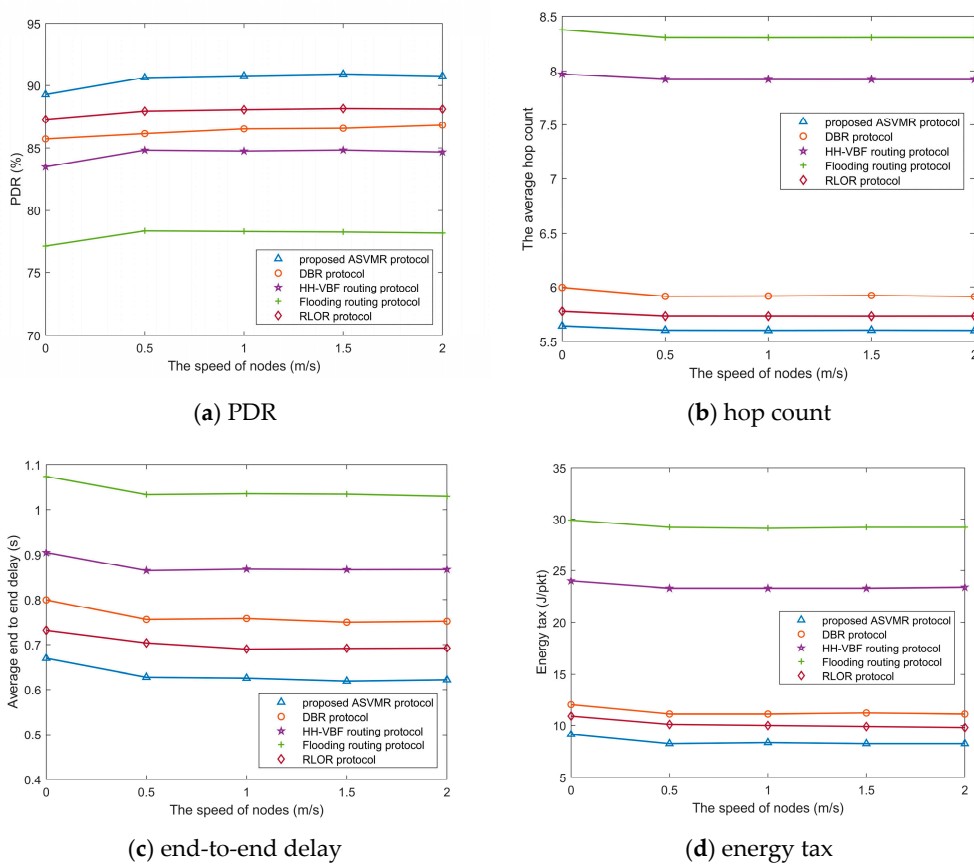

(**a**) PDR

(**b**) hop count

(**c**) end-to-end delay

(**d**) energy tax

**Figure 9.** Impact of the node mobility on the performance.

Therefore, these five routing protocols demonstrate effective handling of sensor node mobility. Moreover, through comparison, the performance of the proposed ASVMR protocol is the best, and the proposed ASVMR protocol is a suitable choice for a mobile UASN.

### 5.3. Impact of Parameter

First, the impact of the discount factor on the performance of the proposed ASVMR protocol is simulated.

Figure 10 shows the impact of the discount factor on the performance of the proposed ASVMR protocol, where the number of sensor nodes is set as 150; the moving velocity of the sensor nodes is set as 0; the discount factor changes from 0.1 to 1; and the routing pipe radius is set as *R*. The PDR, the hop count, the end-to-end delay, and the energy tax are shown in Figure 10a–d, respectively.

From Figure 10, we observe that the magnitude of the discount factor has a certain effect on the performance of the ASVMR protocol. As the value of the discount factor increases, the PDR increases, while the hop count, the end-to-end delay, and the energy consumption decrease.

It is demonstrated in (18) that the discount factor magnitude determines the proportion of the future one-hop decision value to the total decision value in routing decisions. A larger value of $\gamma$ indicates a greater proportion of the one-hop decision value, which results in a better performance of the ASVMR protocol initially. This is because considering the future one-hop decision value in the routing decision makes the decision closer to the global optimal routing than just considering the current decision value.

However, as $\gamma$ further increases, the advantage becomes less apparent because the current state also affects the routing decision. In general, if the computational complexity is not a main concern, incorporating all future states in the decision value calculation would benefit the optimal routing decision.

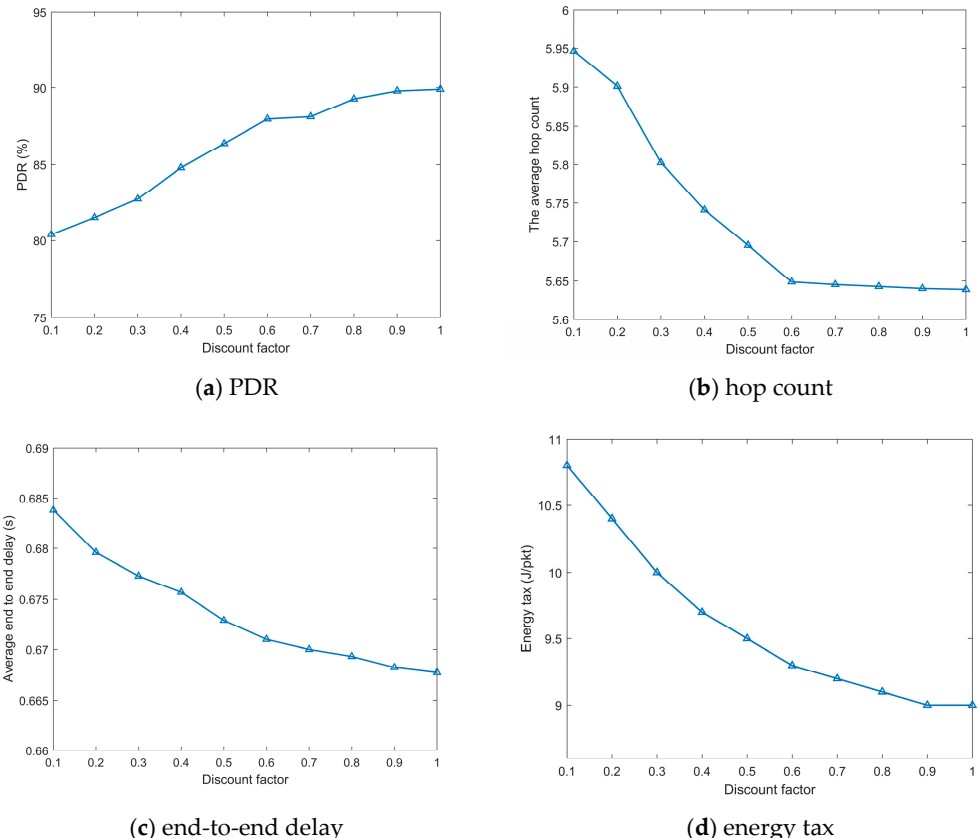

**Figure 10.** Impact of the discount factor on the performance.

Second, the impact of the routing pipe radius on the performance of the proposed ASVMR protocol is simulated.

Figure 11 shows the impact of the routing pipe radius on the performance of the proposed ASVMR protocol, where the number of sensor nodes is set as 150; the moving velocity of the sensor nodes is set as 0; the discount factor is set as 0.8; and the routing pipe radius changes from $R/8$ to $R$. The PDR, the hop count, the end-to-end delay, and the energy tax are shown in Figure 11a–d, respectively.

From Figure 11a, one finds that the PDR of the ASVMR protocol increases with an increase in the routing pipe radius. This is because a larger routing pipe radius can provide more neighboring nodes, which reduces the routing void and lowers the packet loss rate.

From Figure 11b, we observe that as the routing pipe radius increases, the hop count of the ASVMR protocol decreases. The reason for this is the ability of a larger routing pipe radius to provide a greater number of nodes in the candidate forwarding set, thereby increasing the likelihood of discovering a more optimal route.

From Figure 11c,d, we observe that as the routing pipe radius increases, the end-to-end delay and energy tax of the ASVMR protocol decrease. This is because the number of hops from the source node to the destination node decreases, resulting in a reduction in the end-to-end delay and the amount of energy consumed in each packet transmission.

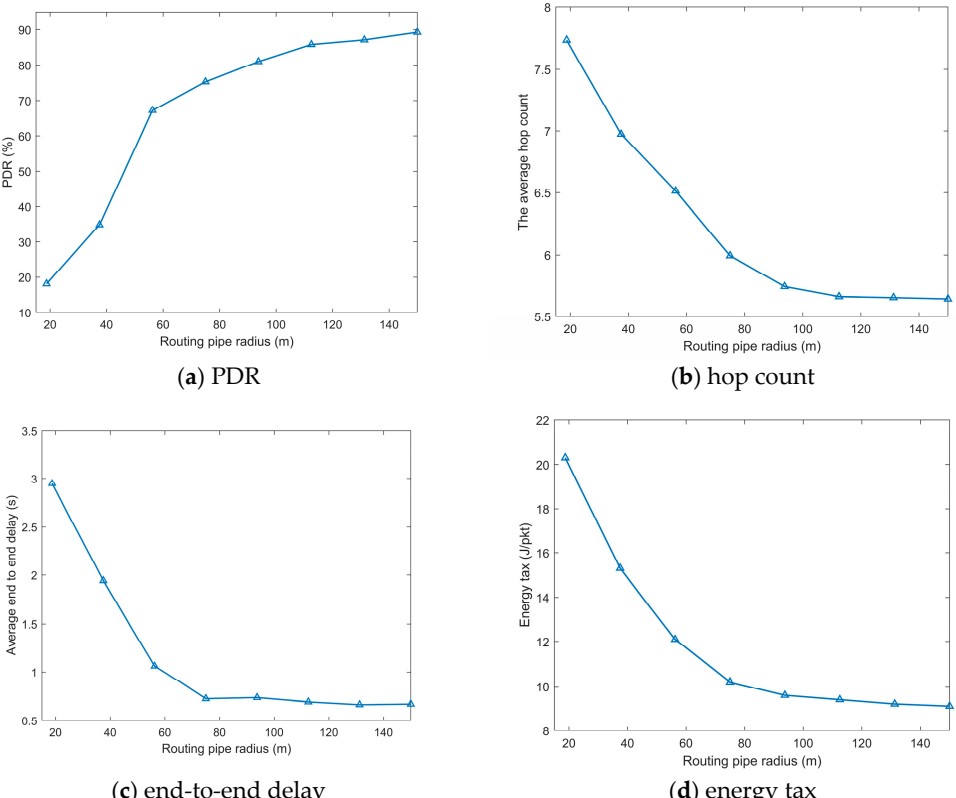

**Figure 11.** Impact of the routing pipe radius on the performance.

Therefore, the routing pipe radius affects the performance of the ASVMR protocol significantly. The increase in the routing pipe radius results in a higher PDR, smaller hop count, and lower end-to-end delay and energy tax in the ASVMR protocol, which indicates that a more optimal path is selected.

## 6. Conclusions

In this paper, we proposed an intelligent routing protocol, the ASVMR protocol, for the UASN. In the proposed protocol, various types of node state information are considered to dynamically optimize the routing path in real time. Specifically, the SVM framework for routing is designed to reduce the end-to-end delay and prolong the network lifetime. The decision-making process for the selection of the next hop incorporates the adaptive routing pipe and future states in order to reduce the energy consumption. Moreover, we introduce a waiting time mechanism and routing void recovery mechanism to improve the PDR and reduce the packet loss in dynamic underwater environments. The simulation results show that the proposed ASVMR protocol performs well in terms of the PDR, the hop count, the end-to-end delay, and the energy tax.

In the future, other parameters in the UASN, such as the transmission rate and the link quality, should be considered to further enhance the accuracy and effectiveness of the routing decision.

**Author Contributions:** Conceptualization, S.Z. and H.C.; methodology, S.Z. and H.C.; software, S.Z.; validation, S.Z., H.C. and L.X.; formal analysis, S.Z. and H.C.; investigation, S.Z.; resources, H.C. and L.X.; data curation, S.Z.; writing—original draft preparation, S.Z.; writing—review and editing, S.Z. and H.C.; visualization, S.Z.; supervision, H.C.; project administration, H.C. and L.X.; funding acquisition, H.C. and L.X. All authors have read and agreed to the published version of the manuscript.

**Funding:** This research was funded by the National Natural Science Foundation of China, grant numbers 42227901 and 62271442, the Science and Technology Department of Zhejiang Province, grant number GG22F019348, and the Natural Science Foundation of Zhejiang Province, grant number LZ23F010006.

**Data Availability Statement:** The data that support the findings of this study are available from the corresponding author upon reasonable request.

**Conflicts of Interest:** The authors declare no conflict of interest.

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
