# Peer review of "ASVMR: Adaptive Support-Vector-Machine-Based Routing Protocol in the Underwater Acoustic Sensor Network for Smart Ocean"

_jmse, doi:10.3390/jmse11091736_

Round 1
Reviewer 1 Report
This paper proposes an adaptive support-vector machine-based routing (ASVMR) protocol for underwater acoustic senor networks that uses a distributed routing approach by dynamically optimizing the routing path in real-time using the four nodes types. It establishes a routing vector from a node sending data to the sink with an appropriate selection of pipe radius based on packet delivery ratio. Futures states of nodes with waiting time are also considering during routing.
My major concern is the logical fallacy of the paper. The proposed ASVMR protocol is compared with the DBR. In DBR a node forwards packets to all of its low depth neighbors in its transmission range in a certain depth threshold. These nodes further forward the packets in the same manner or greedy manner which leads to lot of redundancy packets with excessive energy consumption but the highest packet delivery ratio. However, the proposed protocol sends packets in a restricted pipe/route and still has higher packet delivery ratio than DBR. The simulations need to be performed removing this logical fallacy.
Some other comments worth incorporation are:
In Figure 8, the PDR of the proposed scheme is the highest. That means the proposed scheme delivers the highest number of packets to the destination. As a result, the proposed protocol must consume the highest energy and have possibly the highest delay. But the energy and end-to-end delay of the proposed scheme are lower. This is logical fallacy.
In addition, when a routing pipe is selected for data forwarding nodes in the pipe are frequently chosen that die early due to heavy data traffic and resulting energy consumption and affect the data delivery that results in a rapid data loss in the later round but this behavior is not exhibited in the behavior of PDR.
The proposed protocol is compared with the too old protocols, DBR, HH-VBF and flooding routing. It should be compared with at least one protocol of 2021 or 2022.

English language is fine.
Reviewer 2 Report
jmse-2579310
The article presents an Adaptive SVM-Based Routing Protocol in the Underwater Acoustic Sensor Network for Smart Ocean. The author’s work is timely new and interesting, but the current work seems very limited and needs several improvements. In this regard, some of the suggestions are listed below:
1. First of all, the title should be revised as “ASVMR: Adaptive SVM-Based Routing Protocol in the Underwater Acoustic Sensor Network”
2. There are many grammatical mistakes and typos that need to be corrected with detailed proofreading.
3. The authors have cited a bulk of references at one place to support a minor point, such as [11–18], [11–14], [15–18], and so on. This is just throwing the bundle of articles into the air and after they have hit the ground, which challenging the reader to find something of relevance amongst the resulting mess. It does not help the reader to find more information or evidence. Rather, demonstrate your understanding of the literature by selecting one or two articles that illustrate your arguments well and direct reader to them. It must be avoided.
4. Further latest related work can be added.
5. Comparison of the proposed work with others’ approaches needs to be included properly. Therefore, accurately comparing their model to other underwater Routing Protocol is preferable. If there is no proper comparison, how can the novel reader understand that this model is novel or effective? Therefore, I highly advise comparing your model to the most recent models, particularly those from the last 3-5 years. Moreover, add a separate table for comparison.
6. To know more about underwater communication and routing protocol, the authors can also refer to “A Multi-Layer Cluster Based Energy Efficient Routing Scheme for UWSNs,” IEEE ACCESS”, “Advancements in Neighboring-Based Energy-Efficient Routing Protocol (NBEER) for Underwater Wireless Sensor Networks", Sensors”, “Localization and Detection of Targets in Underwater Wireless Sensor Using Distance and Angle Based Algorithms,” IEEE ACCESS.”
7. Figure 3 resolution is not clear. Please modify and check the remaining figures as well.
8. Check the caption of figure 4, is this correct?
9. How about, if we add Figure 6 caption as “The structure of packet delivery”?
10. Avoid the term “The” at the beginning of every caption.
11. Some abbreviations are repeated such as e packet delivery ratio (PDR), and so on. Check all carefully.
12. The technical depth of the paper could be improved too.
13. The authors claimed that the proposed technique could enhance the accuracy (achieves 99.988%.) in underwater sensor networks, but how?
14. Also, due to the dynamic nature of the underwater environment, how can the nodes be stabilized to get the localization information?
15. Also, is your network affected by the water’s dynamic nature, such as shipping activities and water temperature?
16. Most of the references seems old, it is better to cite more references from the last three to five years.
Extensive editing of English language required
Round 2
Reviewer 1 Report
The authors have incorporated my most concerns.
Some logical fallacies still require a thorough and in-depth analysis (but its ok and the paper can be published in its current form). For instance, the authors said that the proposed scheme sends packets in a pipe that reduces energy consumption and still achieves high packet delivery ratio. This is in contrast to DBR where every node starts sending data packets to all the low depth nodes over all the possible paths in a depth threshold (greedy forwarding). The effect of this is that even multiple copies of a single packets are received at destination that makes the packet delivery ratio of the DBR extremely high. However, the proposed technique sends packets in a restricted pipe that has fairly significantly number of nodes forwarding data packets to the surface compared to DBR that uses nodes in the the entire horizontal plane of the network. Therefore, the two even do not compare in PDR results, the DBR being on the significatly higher side in packet delivery ratio.
Minor proofread required.
Reviewer 2 Report
N/A